# Improving Public Health Policy by Comparing the Public Response during the Start of COVID-19 and Monkeypox on Twitter in Germany: A Mixed Methods Study

**DOI:** 10.3390/vaccines10121985

**Published:** 2022-11-22

**Authors:** Tareq AL-Ahdal, David Coker, Hamzeh Awad, Abdullah Reda, Przemysław Żuratyński, Sahamoddin Khailaie

**Affiliations:** 1Research Associate, Institute of Global Health (HIGH), Heidelberg University, 69117 Heidelberg, Germany; 2Advanced Education Programs, Adjunct Faculty, Fort Hays State University, Hays, KS 67601, USA; 3A/Professor Health Information Management and Health Informatics Research Chair, Higher College of Technology, Abu Dhabi 25026, United Arab Emirates; 4Faculty of Medicine, Al-Azhar University, Cairo 11651, Egypt; 5Division of Medical Rescue, Faculty of Health Sciences with the Institute of Maritime and Tropical Medicine, Medical University of Gdańsk, 80-210 Gdansk, Poland; 6Department of Systems Immunology and Braunschweig Integrated Centre of Systems Biology, Helmholtz Centre for Infection Research, 38124 Braunschweig, Germany

**Keywords:** monkeypox, COVID-19, thematic analysis, Natural Language Processing, unsupervised machine learning, public health policy

## Abstract

Little is known about monkeypox public concerns since its widespread emergence in many countries. Tweets in Germany were examined in the first three months of COVID-19 and monkeypox to examine concerns and issues raised by the public. Understanding views and positions of the public could help to shape future public health campaigns. Few qualitative studies reviewed large datasets, and the results provide the first instance of the public thinking comparing COVID-19 and monkeypox. We retrieved 15,936 tweets from Germany using query words related to both epidemics in the first three months of each one. A sequential explanatory mixed methods research joined a machine learning approach with thematic analysis using a novel rapid tweet analysis protocol. In COVID-19 tweets, there was the selfing construct or feeling part of the emerging narrative of the spread and response. In contrast, during monkeypox, the public considered othering after the fatigue of the COVID-19 response, or an impersonal feeling toward the disease. During monkeypox, coherence and reconceptualization of new and competing information produced a customer rather than a consumer/producer model. Public healthcare policy should reconsider a one-size-fits-all model during information campaigns and produce a strategic approach embedded within a customer model to educate the public about preventative measures and updates. A multidisciplinary approach could prevent and minimize mis/disinformation.

## 1. Introduction

By 11 October 2022, there were more than 620 million COVID-19 cases, with a death toll of over 6.5 million globally [1]. In Germany, 31 million laboratory-confirmed cases of COVID-19 are registered [1]. During the global battle with COVID-19, red flags have been raised concerning the monkeypox situation globally. Moreover, over 65 thousand cases are registered in 106 countries, with 99% of the cases reported from 99 countries with a clean record of this disease. In Germany, the first case of monkeypox was reported on 20 May 2022, and the number increased to 3625 cases by 3 October 2022.

The dynamics of infectious diseases such as SARS-CoV-2 and monkeypox highly depend on social behavior and the extent of social contact. Accurate extraction and tracking of information dissemination and heterogeneity of community perception about the disease is essential to designing, evaluating, and fine-tuning efforts and interventions targeting social connections. In the era of digital life, social media platforms such as Twitter with their extensive user count have proved to be a great source for behavioral and sentiment analysis on various topics [2,3]. Since 2020, COVID-19 has been trending on different social media platforms such as Twitter, and the same is repeating for monkeypox.

This study aims to characterize and compare the early (first three months) social reaction and perception of the two diseases, COVID-19 and monkeypox, based on the data available on Twitter to investigate possible similarities and dependencies between subsequent/concurrent spread. Both COVID-19 and monkeypox were treated as major global health emergencies by the World Health Organization and sparked similar levels of fear and uncertainty; many laypersons thought both could be the next pandemic [4,5]. For many, the public perception indicated that monkeypox could be a continuation of COVID-19 policies.

To this end, following a retrospective historical analysis before COVID-19 and monkeypox preventative measures were developed, we employed machine learning methods with thematic analysis of Twitter data to track meaningful discussions and sentiments during the first three months of COVID-19 and monkeypox. Many researchers document problems and provide prescriptions to improve public policy but do not capture public sentiment [6,7]. There is a gap in understanding beyond surveys, network analysis, and quantitative research [8,9,10], which could lead to enhanced understanding by qualitative analysis. These analyses can inform health authorities on ways to assess and regulate the situation by disseminating information that alleviates distrust and concerns.

Previous studies showed the promising use of Twitter in the “infodemiology” studies related to the spread of infectious diseases. A study by Xue et al. [11] showed the potential of machine learning with Twitter data in enabling research related to public health. They stated that pandemic-related fear, stigma, and health concerns are already evident and influencing public trust during epidemiological waves. Boon-Itt and Skunkan showed the usefulness of sentiment analysis and topic modeling in producing information about the trends in the discussion of COVID-19 [2]. They emphasized that Twitter is a good source of information for understanding public concerns and awareness. In addition, understanding the situation and sharing the results with decision-makers could help the health departments select information to alleviate public concerns about the disease.

## 2. Materials and Methods

### 2.1. Sample

Our sampling approach was purposive, as shown in Figure 1. Tweets related to COVID-19 between 1 January 2020 and 31 March 2020 and monkeypox between 1 May 2022 and 31 July 2022 were selected. We recruited tweets in the German language and restricted our search to Germany only. Tweets which were not about COVID-19 or monkeypox or were in another language were not selected.

### 2.2. Methodology

Sequential explanatory mixed methods research was conducted, as shown in Figure 2. Latent Dirichlet Allocation (LDA) is a machine learning approach that helps researchers to analyze contextual data such as Twitter messages. This algorithm produces frequently mentioned pairs of words, the teams of words that co-occur, and the latent topic and their distribution. Some studies have also shown the feasibility of using LDA for patterns and theme identification [11]. After collecting the data, we followed well-known and validated procedures. We prepared the raw data by Natural Language Processing ‘NLP’ and Natural Language Toolkit ‘NLTK’ libraries specifying that it is for the German language; we removed all the symbols, we removed the hashtags, the user handles, the multiple spaces, the URLs, the punctuations, the HTML tags and the numbers, and finally tokenized the words and obtained the stems [12].

After using (NLP) and (LDA), a qualitative approach using thematic analysis examined the COVID-19 and monkeypox tweets [13,14,15,16,17]. Qualitative research does not rely on variables, predictive models, or hypotheses, but examines data to explain the how, the what, and the why behind peoples’ thinking beyond what a survey or other quantitative measures can capture [18,19]. Developing an understanding of the way in which people interpret a phenomenon is the goal: “Like healthcare, education involves complex human interactions that can rarely be studied or explained in simple terms. Complex educational situations demand complex understanding; thus, the scope of educational research can be extended by the use of qualitative methods” [20] (p. 1).

A novel rapid tweet analysis protocol (RTAP), Table 1, was employed within the thematic framework. First, each tweet sample was read and annotated in pre-binning to develop nascent categories. All tweets have a minimum of three categories: noninformative, advertisements, and humor. Then, each tweet sample was analyzed page by page and placed into bins, or the nascent and/or new categories were developed; keywords from the LDA helped to focus analysis to ensure all significant trends were covered as well as develop specific categorical descriptions. Since there was multivocality with limited interaction, major trends were reported; there were minor trends and ambiguity, as many messages were short and out of context. Post-binning involved reconciling all categories for consistency and the development of themes and metathemes. Finally, each tweet sample was analyzed with compare and contrast and written in narrative form.

### 2.3. Data Analysis

The LDA machine learning method was utilized for classifying the words into topics and calculating the coherence score of the result. The following query words were used: COVID-19, #COVID-19, Corona cases, Coronavirus, Monkeypox, #Monkeypox, Monkeypox virus, Monkeypox cases. The data was retrieved from Twitter API academic access using the Tweepy library in Python programming language. The collected information for COVID-19 comprised 8532 tweets. After the removal of duplicated tweets, 5349 remained. For monkeypox, the total collected tweets were 7404. After the removal of the repeated tweets, 4552 remained.

Then, thematic analysis using a rapid tweet analysis protocol was employed to examine and compare and contrast the COVID-19 and monkeypox sample. Using Google Translate, all tweets were translated into English. Thematic analysis was conducted to analyze data [5,7,8]. First, there was a review of the machine learning results, and then the data were read, annotated, and queried to develop pre-binning, or likely categories to quickly analyze data. Tweets could be split and placed into multiple categories. Using Microsoft Word and Excel, the following coding schema was used in each sample using bins, or categories: in vivo, descriptive, elements, dimensions, ahas, and memoing. Keyword analysis assisted in a second examination. Post-binning involved several steps. Themes were formed by examining repetition, opposites, absences, and degrees. Themes and metathemes were constructed, defined, and referenced. Before reporting, there was a search for divergences and negative cases. The results of each sample were examined side-by-side for similarities and differences. Multivocality existed and meant many stubs and nuances, so only the most prevalent and relevant themes were presented. In our approach, two members reviewed the codes and categorized them into specific themes and meta-themes.

## 3. Results

LDA was initiated by proposing 12 topics and measuring their coherence; we then calculated the coherence score and U_Mass of different topic numbers from 7–20 topics to find the best topic number; it was about 0.5330 for Topic 17. We measured its U_Mass, and it was about −15.94319. From that, we can see that the best coherence and U_Mass scores were with the number 17, which was the least number of topics possible with high coherence.

Further analysis of the document-term matrix was performed, which was used to obtain the distributions of 17 topics of the top 30 words. The results of 17 salient issues and the most popular pairs of words (bigrams) within each topic are presented in Appendix A. For example, Topic 7 for COVID-19 had the highest distribution (10.4%) among all 17 common latent issues. The bigrams associated with Topic 7 included “responsibility”, “home”, “problem”, “government”, and “probably”. These pairs of words frequently co-occurred, and the LDA model assigned them to the same topic. For instance, Topic 2 for monkeypox dataset had the highest distribution (10.5%) among all 17 common latent topics, and the bigrams associated with this topic “Monkeypox”, “Vaccination”, “recommendation”, and “vaccine dose”. Word clouds, in Figure 3, show the differences in tweets between COVID-19 and monkeypox.

Using thematic analysis with the rapid tweet analysis protocol, major and minor themes were developed comparing COVID-19 and monkeypox (see Table 2). Three major themes emerged in the analysis of tweets during the beginning of the COVID-19 pandemic: emerging personal effects of COVID-19, the spread of the disease, and commentary on the outlook of the crisis. The spread and personal effects were intricately woven together; reports of global spread to nearby countries and eventually one’s locality permeated much of the discussions. The numbers gradually increased as cases became more real by becoming local, with frequent reports of the cases and deaths increasing dramatically with each passing week. Severe illness and death were of great concern to many. As the spread increased, personal effects became negative and included quarantine/stay at home (“stay at home”, “keep a safe distance”, and “get together” #online, etc.), closing of businesses and sporting events (“bans all major events”, “cancel large events”, and “won’t fill football stadiums until next year”, etc.), and hoarding (“products sold out”, “no toilet paper”, and “#hamsterkauf”, etc.). Debates ensued about the prevalence, severity, and reactions. Some felt COVID-19 was exaggerated (“8 deaths, panic”, “hardly worse than the ordinary flu”, “don’t let myself be scared”, etc.), while others felt gloom and doom (“brink of catastrophe”, “eerie picture like out of a horror movie”, “how deadly… far too much”, and “seriousness… makes you sad and speechless”, etc.), and others yet mentioned hope (“all the best” and “humanity and solidarity”, etc.) among the many voices.

Multivocality meant that many voices could not quickly coalesce into one theme. Some minor themes included humor about the disease, the spread compared to prior pandemics, and the actions of politicians (e.g., “What does the #coronavirus and the associated #hamster purchases actually bring to someone who doesn’t like pasta. only advantages, right?”, etc.); advertisements about products to protect the consumer, and fake news, both spreading and recognizing (“faktencheck has now developed a new reporting system”, “@bild needs the circulation and is happy about populist headlines”, etc.). From blaming China to comparisons to the flu, tweets ranged from making jokes to calls for calm and reduction of irresponsible behavior. Many expressed gloom while others saw exaggeration. A factor which tied all issues together was COVID-19′s increased spread until there were personal connections and restrictions, which caused a ripple in the way most people lived.

Monkeypox analysis produced four significant themes: development, use, and questions about vaccinations; doubts and questions about the severity, information, or response; reports of global spread, with tweets reporting cases with a slight skew toward Germany, and the political actions in response to monkeypox, with Lauterbach and the World Health Organization (WHO) being prominent. Vaccinations were questioned from the perspective of efficacy, implementation of programs, and viability of past programs. For example, there were questions about the smallpox vaccination (“the good thing about monkey pox is that all boomers are already vaccinated”, “biological weapons smallpox”, etc.), side effects of shingles and the public concern of Dr. Wodarg (“vaccination side effect is sold as monkeypox”, “solves the mystery of shingles/monkey pox”, etc.), and who/when to vaccinate (“risk groups and children”, “will be ready”, and “ordering 8000”, etc.). Many tweets cast significant doubt, with updated information seen as possibly conspiratorial: bioweapons, an extension of political control, and exaggerated danger, to name a few (“…self-limiting and harmless”, “… msm are the focus of the utbreak [sic]”, “affenpocken…a new edition from 2018? just ridiculous you so-called journalists”, and “monkey pox: google -& biological weapons smallpox… ”, etc.).

The spread of the disease was tweeted throughout, but the reports were much more infrequent than those of COVID-19 and had a much greater focus on the global spread. For many, the virus did not manifest into a personal concern, either directly or indirectly. Tweeters seemed to have COVID-19 fatigue, seeing politicians in a much more negative light than in the case of COVID-19. The WHO (“the #who declares the monkeypox emergency. against the vote of our own experts!”, “the #who declares an emergency… controversial decision”, and “who has called for a sex shutdown”, etc.) and especially Lauterbach (“karl_lauterbach don’t you still have enough panic to spread with corona, influenza and monkeypox?”, “#lauterbach resigns immediately”, “many unhappy”, and “he cheerfully carries on as if nothing happened”, etc.) received the brunt of the criticism.

Minor themes were common within a diverse population. Humor (“Because of the increasingly contagious #corona… everyone must be entitled to it!”), homosexuals (“ministry of health’s statement with stigmatizing catastrophe communication like in the 80s”, “very honest text by noam about monkey pox, gay sex and the failure of berlin politics”, “528 #monkeypox cases in 16 countries: 98% were gay or bi men… 41% with hiv”, and “berlin is full of homos, hence the high number of infections”, etc.), and deaths (few reported, mortality rate listed as low or from preexisting conditions, etc.) were the three minor themes. Humor ranged from crude to sarcastic. Since gay men were a group identified as higher-risk, tweets transformed from stigmatization to homophobia. Lastly, deaths were far less of a concern than in the case of COVID-19.

Many researchers reported the emotions in tweets, but many tweets were indecipherable. With only 160 characters and lacking verbal and nonverbal cues, one often could not be certain if someone was angry or happy or if the speech was sarcastic or severe. Many tweets were also taken out of context, so the emotional aims were ambiguous in many tweets.

### Meta-Themes and Comparison

As shown in Table 3**,** there were two major differences between the two responses. Selfing described the meta-theme for COVID-19: some saw no imminent danger ahead, others believed the situation was minor, and others yet felt there was a crisis. Selfing describes the process of the tweets; the collectivization of the experience was deeply felt, regardless of agreement about the danger or types of response. Monkeypox could be surmised as othering: other people in other places suffered, resulting in a full recovery. Fatigued by the COVID-19 answer and the failure of monkeypox to make a direct or indirect impact on most tweeters’ lives meant that many felt free to question and doubt the media and political leaders. Most felt the vaccine was a cure-all for a disease of little importance.

Instead of a consumer model during COVID-19, tweeters aligned closer to a customer model, where the participants searched for coherence amidst reconceptualization. With far less media coverage and risks seemingly lower, the monkeypox epidemic was experienced and reconceptualized during a “crisis within a crisis”. There was more debate and mis/disinformation during the beginning of the monkeypox tweets than COVID-19. Perceived insularity from monkeypox, real and imagined (lower mortality rate and a vaccine), caused many people to shop and compare data in a marketplace of competing ideas, often regardless of trust and veracity.

## 4. Discussion

There were few qualitative analyses, but the comparisons converged and diverged from the current findings. With monkeypox, there are no satisfactory comparisons. Many researched post hoc the results of containment policies, [21] vaccinations of COVID-19, [22,23,24,25] and other preventative measures, [22,26,27] but little research offers a historical perspective in vivo of the public’s emerging conceptualization and personal response at the commencement of public health crises.

Two direct comparisons for COVID-19 have been previously reported [28,29]. Analyzing Arabic tweets, responses to protect oneself and the public and the spread of the disease dominated tweets [29], which was similar to the German dataset analyzed here. Thelwall and Thelwall diverged in their findings, with lockdown, humor/attitude, the political, and safety measures being prominent [28].

Machine learning offered comparisons. Local spread of COVID-19, as in the current study, was of interest early on [30]. Iranians tweeted about quarantine, the spread of the disease, and the Iranian regime; others, analyzing English, Chinese, and Japanese tweets, found that spread and deaths at the top of the list were similar, though economic impact became more prevalent over time [31,32]. Xue et al. [11] produced similar findings regarding the spread/deaths, reactions/prevention, and political actions. Other researchers found similar sentiment analysis to the current study [33,34].

Much less is known about monkeypox in the Twitterverse, and what was researched lacked any qualitative findings. Early analysis suggested an abnormally high misinformation rate and use of tweets to track outbreaks [35,36]. Machine learning found significant overlap with the current study’s findings: the homosexual community concerns, spread and death causing much less concern, severity of the disease, safety, lack of faith in public institutions, nature of vaccinations, and if COVID-19 was repeating, among other topics [37,38]. Compared to COVID-19, monkeypox remained largely impersonal for most people.

Within selfing and othering, public messaging should be considered within a customer model instead of a producer/consumer model. The concept of bending within narratives or forming issues within one’s images and life should be considered [5,39]. COVID-19 impacted large swaths of the world’s population negatively in all areas of life and caused mental, social, and emotional problems [40,41,42,43], producing pandemic fatigue by the start of monkeypox, with uncertainty and increasing media and government distrust [44,45]. The fatigue caused feelings of doubt about science itself because being inundated with redundant information can cause doubt [46,47]. The expediency of preventative measures for COVID-19 failed to materialize [48], causing a shift in processing monkeypox.

A customer model means members of the public buy and own knowledge that fits their worldview, as opposed to scientific and general health knowledge (producer) being served (consumed) because of who and what was said. The divergence contrasts previous models, which ignore important ecological variables and lack strategic implementation. Social media develops an emotional and personal connection with public health emergencies [49]. Scientists and public health agencies often fail to realize that updated findings are often perceived as mis/disinformation to customers who believe contradictory findings are evidence of a lack of credibility. Laypersons need to know that science relies on all results as tentative, which can also be a struggle within the scientific community [50].

Continued crises and quickly developing information should be considered within a strategic implementation framework, and messaging of results cannot be reduced to a simple, one-size-fits-all model [51,52,53]. A customer model considers the factors and situational complexity while analyzing the decision-making processes which produce biases [51,54,55]. Using cognitive sciences, healthcare campaign models currently proposed lack specificity and suffer from ecological validity because decision-making processes defy technical-rational solutions.

We propose a multidisciplinary framework using the knowledge of business and education and journalism, etc., using the protocol in a rapid, ongoing fashion: (1.) Consideration of new information within the context of previous data to gauge receptivity; (2.) Rapid prototyping and testing new messages with focus groups and surveys; (3.) Scoring the sale and purchase (belief) of public health knowledge. With social media, combating mis/disinformation often rooted in the verisimilitude of scientifically accurate information [35] means public health agencies must consider the fit within the current marketplace of ideas.

A key problem is the fact that time precludes rapid analysis; machine learning is limited in offering explanatory value. There is a dearth of qualitative research in analysis of large-scale social media data collections. While machine learning can identify negative information or misinformation [56], qualitative research tells the story behind keywords and sentiments. The rapid tweet analysis protocol could assist researchers in developing a valid and reliable framework by applying the concept of statistical bins to qualitative research; when coupled with descriptors and subcategories, major trends and issues can be revealed.

## 5. Conclusions

The strength of the study is a mixed methods design; a qualitative examination added needed context to machine learning results. A limitation of the study is the fact that only one country’s response to COVID-19 and monkeypox was considered. The data might not be representative of all regions in Germany. Tweets can be short and decontextualized, rendering many noninformative and lacking emotional resonance.

Public health responses need to be timely, accurate, and constantly updated to effectively respond to the emergence of health emergencies [57]. Social media is a major factor in public health responses and campaigns [58], but existing studies on vaccine hesitancy and other counterproductive actions by the public lack insight into the way to implement their recommendations [59]. The results of this mixed method study using data from Twitter to compare the discussions between two public health threats have shown that the public during monkeypox was unconcerned about the preventive measures and produced a very different response than in the case of COVID-19. Monitoring mechanisms which impact public opinion and reactions to public health concerns could improve messaging and marketing of public health policies.

Benati and Coccia (2022) [60] point out the value of good governance in improving vaccination programs, and the current study suggests understanding the factors of public concern could improve prevention and mitigation efforts. Rapid qualitative analysis can tell the why, the how, and the what of public opinion, which cannot be easily captured with quantitative techniques which predominate in the science fields. We recommend that health institutions share more information regarding the disease risk, which connects with the public personally, and develop a customer-based campaign to test and target messaging. A strategic, multidisciplinary approach and monitoring of the public response and resistance to messaging could improve prevention and mitigation efforts. The rapid tweet analysis protocol could assist teams in efficiently analyzing large datasets to support public health campaigns.

## Figures and Tables

**Figure 1 vaccines-10-01985-f001:**
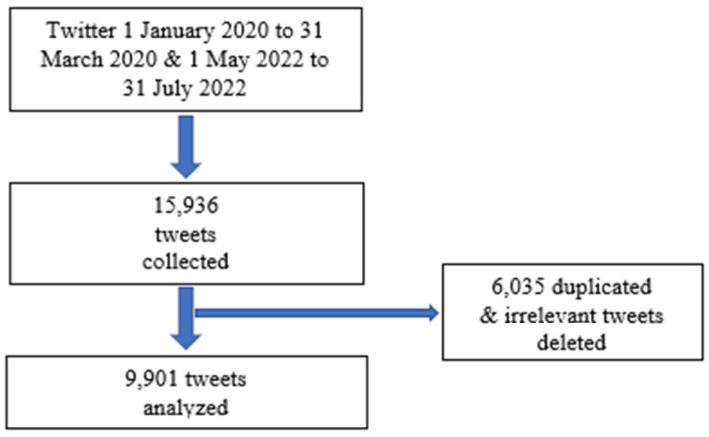
Process to select sample tweets.

**Figure 2 vaccines-10-01985-f002:**
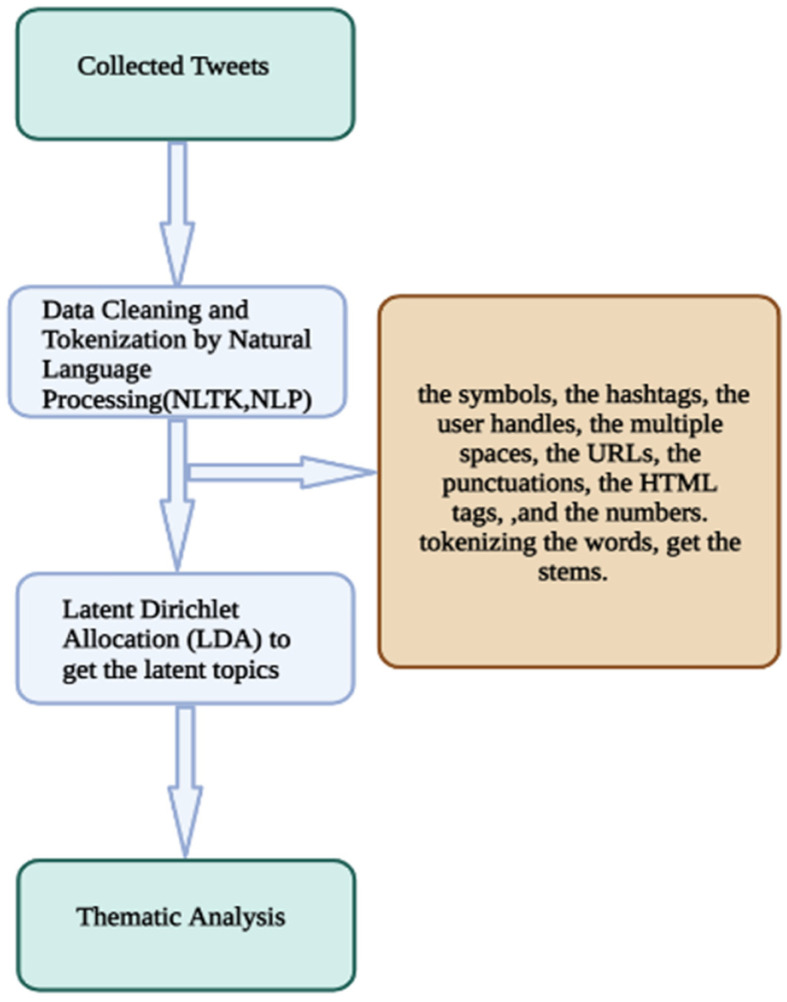
Sequential explanatory mixed methods process.

**Figure 3 vaccines-10-01985-f003:**
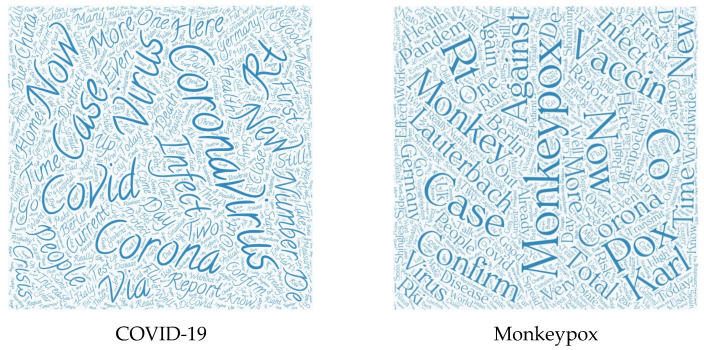
Word cloud analysis of COVID-19 and monkeypox.

**Table 1 vaccines-10-01985-t001:** Rapid tweet analysis protocol.

Step	Process
1. Pre-binning	Developing nascent categories by reading, annotating, applying machine learning, and keyword analysis
2. Binning	Coding and categorization of tweets into bins by topic and trends, checked by machine learning
3. Thematizing	Sorting and checking of categories into elements, dimensions, and themes
4. Post-binning	Checking and reconciling themes within each theme, across themes, and with the data
5. Reporting	Comparing and contrasting results and producing a narrative

**Table 2 vaccines-10-01985-t002:** Comparison of themes between COVID-19 and monkeypox.

Type	COVID-19	Monkeypox
Major themes	personal effects local spread outlook on impending crisis	vaccinations questioning science global spread political actions
Minor themes	humor advertisements past crises	humor homosexual spread/concerns low death rate

**Table 3 vaccines-10-01985-t003:** This table shows the comparison for meta-themes for COVID-19 and monkeypox.

Topics	COVID-19: Selfing	Monkeypox: Othering
Reporting Cases	Proximity. Cases are reported worldwide but strongly focus on one’s country and community.	Distal. Cases focused on global issues, with one’s country and community a minor issue.
Deaths	Alarm. The death rate increased, and it came closer to home.	Rare. Deaths were infrequent and marginalized as a concern.
Protective Measures	Personal Ripple. The evolution of protective measures (quarantines, stay at home orders, social distancing, etc.) affected everyone.	In the News. There were no protective measures, and monkeypox was elsewhere. Vaccines meant there was a low risk.

## Data Availability

Data are available upon request from the corresponding author.

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
