# Peer review of "Improving Public Health Policy by Comparing the Public Response during the Start of COVID-19 and Monkeypox on Twitter in Germany: A Mixed Methods Study"

_vaccines, 2022, doi:10.3390/vaccines10121985_

Round 1

Reviewer 1 Report

The article by Tareq AL-Ahdal et al. investigates the public response during the first three months of two major health events (COVID-19 and Monkeypox).

The title reflects the article’s content.

The results and observations are relevant to the problem posed and provide proof for the consistency of literature data.

The work is original and scientifically reliable and the methods are appropriate and adequately described.

The article is quite well presented, although with few tables and figures.

The speculations made are reasonable and all the interpretations are warranted by the data gathered.

The message is transmitted clearly and the overall design of the study is good, the undertaken research is properly described and the conditions are well defined.

The English is understandable to the reader.

Author Response

Thank you for your kind words and insightful response.

Reviewer 2 Report

In this manuscript, the authors aimed to characterize and compare the early (first three months) social reaction and perception of the two COVID-19 and monkeypox based on the data available on Twitter. The idea of this study is novel and might be interesting to readers. Nevertheless, this manuscript suffers from several major limitations.  

The biggest limitation is that the results of the study are not clearly presented. It seems there are two “Figure 1” in the manuscript. In addition, there is only one table presented in the paper but it is labeled as “table 2”. The core information of results of an academic study should be expressed through tables and figures but not large trunk of texts in the results part. This will make the paper less attractive for potential readers and make it very difficult for readers to understand.

Methods part is also not very clear. The authors mentioned that they have implemented a machine learning based method to conduct their analysis. But no detail information is provided. What is the training set? What is the validation set? How about the performance of their models? What algorithms they utilized for the machine learning methods?

The language is another big problem. Multiple typos can be identified from the manuscript. The figure legend of this manuscript is like “Figure 1. This is a figure for data Retrieving”. That is just so colloquial. That is wired to write something like that in an academic paper. Extensive professional language editing are needed.

Author Response

In this manuscript, the authors aimed to characterize and compare the early (first three months) social reaction and perception of the two COVID-19 and monkeypox based on the data available on Twitter. The idea of this study is novel and might be interesting to readers. Nevertheless, this manuscript suffers from several major limitations.  

Thank you for the kind words. We did not identify any comparative studies, which can add insight to public health policies.

The biggest limitation is that the results of the study are not clearly presented. It seems there are two “Figure 1” in the manuscript. In addition, there is only one table presented in the paper but it is labeled as “table 2”. The core information of results of an academic study should be expressed through tables and figures but not large trunk of texts in the results part. This will make the paper less attractive for potential readers and make it very difficult for readers to understand.

We added figures, tables, and explanations. We hope we strengthened the paper.

Methods part is also not very clear. The authors mentioned that they have implemented a machine learning based method to conduct their analysis. But no detail information is provided. What is the training set? What is the validation set? How about the performance of their models? What algorithms they utilized for the machine learning methods?

We clarified the sections and structured them similar to other articles in Vaccine.

The language is another big problem. Multiple typos can be identified from the manuscript. The figure legend of this manuscript is like “Figure 1. This is a figure for data Retrieving”. That is just so colloquial. That is wired to write something like that in an academic paper. Extensive professional language editing are needed.

We agree there needed to be updates. Thank you for your assistance.

Reviewer 3 Report

Measuring the public response during the first three months of COVID-19 and Monkeypox by using data from Twitter in Germany: A sequential explanatory mixed methods study

The topics of this paper are  interesting. 

Frankly the comparison between COVID-19 and Monkeypox is not clear. COVID-19 is a pandemic, Monkeypox has not had the same diffusion and impact. 

The structure and content must be revised, and results have to be  better explained by authors before to be reconsidered for publication.

Title has to be shorter. 

Abstract has to clarify the goal, results and  health and social implications to improve prevention and preparedness to cope with next pandemics similar. 

Introduction has to better clarify the research questions of this study and  describe the different sources of transmission dynamics of COVID-19 and risk factors in society, as well as the role of non-pharmaceutical interventions and vaccinations that have affected social reaction and perception of high impact in COVID-19 compared to monkeypox (also considering the scope of the journal). After that they can focus on these topics  (See suggested readings that must be all read and used in the text). 

Methods of this study is not clear. They have to be restructured with three sections :

•    Sample and data

•    Measures of variables (to be inserted)

•    Models and Data analysis procedure. 

There are two figures with title of figure 1 in methods. This is confusing. .

Figures have to be clarified to indicate what they represent. 

Results. 

Authors have to avoid subheading that create fragmentation and confusion. If necessary, can use bullet points. 

In results there is table 2 and not table 1…This is confusing…and low accuracy. 

A comparative analysis of social reaction and perception between these two infectious diseases should be better synthetized in a table to show really the differences and possible similarities and dependencies between subsequent/concurrent pandemics.

However results are obvious and interpretation is vague. Authors have to explain that strict containment policies and pressure to vaccinations for COVID-19 with news and newspapers, etc. have generated a high social pressure and high social reactions compared to monkeypox. This has to be explained in discussion considering the literature suggested.

Discussion has to show what this study adds compared to previous results in term of prediction and crisis management as well as how to improve governance  to face next  pandemics and improve communication. 

Conclusion so short is useless. It has to be extended and has not to be a summary, but authors have to focus on manifold limitations of this study and provide suggestions of health, crisis management  and social policy to improve healthcare sector, prevention, social communication,  and preparedness to cope with next pandemics based on new infectious disease similar to COVID-19. 

Overall, then, the paper is interesting. Theoretical framework is weak, and some results create confusion… 

The comparison between these two infectious diseases is not consistent because COVID-19 is a big pandemic, Monkeypox has had a low impact and of course emerge differences that have to be better synthetized in a table. These aspects have to be accurately discussed also in terms of limitations of this study. Additionally for COVID-19 there was lockdown and other policy restrictions for Monkeypox not and this has to be also explained to justify the different perceptions and  social reaction of mental health.  Moreover, , structure of the paper has to be improved; study design, discussion and presentation of results have to be clarified using suggested comments.

If the paper is improved considering that a low relation with the scope of the journal,  by using strictly all comments that I will in-depth verify,  maybe it can be considered. 

Suggested readings of relevant papers that have to be read and all inserted in the text and references.

Qorib, M., Oladunni, T., Denis, M., Ososanya, E., Cotae, P. 2023Covid-19 vaccine hesitancy: Text mining, sentiment analysis and machine learning on COVID-19 vaccination Twitter dataset. Expert Systems with Applications212,118715

Chowdhury T., Chowdhury H., Bontempi E., Coccia M., Masrur H., Sait S. M., Senjyu T. 2022. Are mega-events super spreaders of infectious diseases similar to COVID-19? A look into Tokyo 2020 Olympics and Paralympics to improve preparedness of next international events. Environmental Science and Pollution Research, https://doi.org/10.1007/s11356-022-22660-2

Arbane, M., Benlamri, R., Brik, Y., Alahmar, A.D. 2023Social media-based COVID-19 sentiment classification model using Bi-LSTM. Expert Systems with Applications212,118710

Núñez-Delgado A., Bontempi E., Coccia M., Kumar M., Farkas K., Domingo, J. L. 2021. SARS-CoV-2 and other pathogenic microorganisms in the environment, Environmental Research, Volume 201, n. 111606, https://doi.org/10.1016/j.envres.2021.111606.

Shoaib, H.M. 2023The Influence of Visual Risk Communication on Community During the COVID-19 Pandemic: An Investigation of Twitter Platform. Studies in Systems, Decision and Control216, pp. 349-364

Benati I., Coccia M. 2022. Global analysis of timely COVID-19 vaccinations: Improving governance to reinforce response policies for pandemic crises. International Journal of Health Governance. https://doi.org/10.1108/IJHG-07-2021-0072

Semeraro, A., Vilella, S., Ruffo, G., Stella, M. 2022Emotional profiling and cognitive networks unravel how mainstream and alternative press framed AstraZeneca, Pfizer and COVID-19 vaccination campaigns. Scientific Reports12(1),14445

Coccia M. 2021. Comparative Critical Decisions in Management. In: Farazmand A. (eds), Global Encyclopedia of Public Administration, Public Policy, and Governance. Springer Nature, Cham. https://doi.org/10.1007/978-3-319-31816-5_3969-1

Alassad, M., Agarwal, N. 2022Contextualizing focal structure analysis in social networks. Social Network Analysis and Mining12(1),103

Coccia M. 2021. The relation between length of lockdown, numbers of infected people and deaths of COVID-19, and economic growth of countries: Lessons learned to cope with future pandemics similar to COVID-19. Science of The Total Environment, vol. 775, article number 145801, https://doi.org/10.1016/j.scitotenv.2021.145801

Stracqualursi, L., Agati, P. 2022Tweet topics and sentiments relating to distance learning among Italian Twitter users, Scientific Reports12(1),9163

Coccia, M. 2021. Pandemic Prevention: Lessons from COVID-19. Encyclopedia, vol. 1, n. 2, pp.  433-444. doi: 10.3390/encyclopedia1020036

Ogbuokiri, B., Ahmadi, A., Bragazzi, N.L., (...), Asgary, A., Kong, J. 2022Public sentiments toward COVID-19 vaccines in South African cities: An analysis of Twitter posts. Frontiers in Public Health, 10,987376

Coccia M. 2022. Optimal levels of vaccination to reduce COVID-19 infected individuals and deaths: A global analysis. Environmental Research, vol. 204, Part C, March 2022, Article number 112314, https://doi.org/10.1016/j.envres.2021.112314

Jahanbin, K., Jokar, M., Rahmanian, V. 2022Using twitter and web news mining to predict the monkeypox outbreak. Asian Pacific Journal of Tropical Medicine15(5), pp. 236-238

Coccia M. 2022. COVID-19 Vaccination is not a Sufficient Public Policy to face Crisis Management of next Pandemic Threats. Public Organization Review, https://doi.org/10.1007/s11115-022-00661-6

Ortiz-Martínez, Y., Sarmiento, J., Bonilla-Aldana, D.K., Rodríguez-Morales, A.J. 2022Monkeypox goes viral: measuring the misinformation outbreak on Twitter. Journal of Infection in Developing Countries16(7), pp. 1218-1220

Coccia M. 2022. Effects of strict containment policies on COVID-19 pandemic crisis: lessons to cope with next pandemic impacts. Environmental Science and Pollution Research, DOI: 10.1007/s11356-022-22024-w, https://doi.org/10.1007/s11356-022-22024-w

Liu, L., Fu, Y. 2022Study on the mechanism of public attention to a major event: The outbreak of COVID-19 in China. Sustainable Cities and Society81,103811

Author Response

The topics of this paper are interesting. 

Thank you. We believe the topic is of interest and relevance to developing public health campaigns.

Frankly the comparison between COVID-19 and Monkeypox is not clear. COVID-19 is a pandemic, Monkeypox has not had the same diffusion and impact. 

Significant update, as the initial public perception viewed monkeypox as a pandemic. The WHO declaring monkeypox a public health emergency stimulated concern among the public.

The structure and content must be revised, and results have to be better explained by authors before to be reconsidered for publication.

 All updates as requested.

Title has to be shorter. 

Title was revised following the length and style of Coccia’s articles.

Abstract has to clarify the goal, results and  health and social implications to improve prevention and preparedness to cope with next pandemics similar. 

 Updated and clarified.

Introduction has to better clarify the research questions of this study and  describe the different sources of transmission dynamics of COVID-19 and risk factors in society, as well as the role of non-pharmaceutical interventions and vaccinations that have affected social reaction and perception of high impact in COVID-19 compared to monkeypox (also considering the scope of the journal). After that they can focus on these topics  (See suggested readings that must be all read and used in the text). 

 Introduction was updated and clarified.

Methods of this study is not clear. They have to be restructured with three sections :

•    Sample and data

•    Measures of variables (to be inserted)

•    Models and Data analysis procedure. 

The sections were clarified and improved. Since the study used qualitative analysis, there are not variables (with an exploratory qualitative analysis, one mines the data to form the variables post priori).

There are two figures with title of figure 1 in methods. This is confusing.

Thank you. As often happens during revisions, figures/tables often misnamed.

Figures have to be clarified to indicate what they represent. 

Figures updated.

Results. 

Authors have to avoid subheading that create fragmentation and confusion. If necessary, can use bullet points. 

All Vaccine articles have a style guide which include subheadings of results; bullets are not an acceptable style in MDPI. Furthermore, all mixed methods studies in Vaccine have the style we adopted. If the editors change the style, we will be happy to make any edits requested.

In results there is table 2 and not table 1…This is confusing…and low accuracy. 

Corrected.

A comparative analysis of social reaction and perception between these two infectious diseases should be better synthetized in a table to show really the differences and possible similarities and dependencies between subsequent/concurrent pandemics.

Many tables and figures were added.

However, results are obvious and interpretation is vague. Authors have to explain that strict containment policies and pressure to vaccinations for COVID-19 with news and newspapers, etc. have generated a high social pressure and high social reactions compared to monkeypox. This has to be explained in discussion considering the literature suggested.

While this issue was addressed in the discussion, the strict containment policies and pressures are ahistorical, i.e., we clarified this study is a retrospective historical study. As such, within the data analysis, none of these measures had yet happened. Hence, there is no data in our study addressing any of these issues because all these were future events compared to our tweet analysis of the first three months of each epidemics.

Discussion has to show what this study adds compared to previous results in term of prediction and crisis management as well as how to improve governance to face next  pandemics and improve communication. 

Updated and clarified.

Conclusion so short is useless. It has to be extended and has not to be a summary, but authors have to focus on manifold limitations of this study and provide suggestions of health, crisis management  and social policy to improve healthcare sector, prevention, social communication,  and preparedness to cope with next pandemics based on new infectious disease similar to COVID-19. 

Conclusion updated and modelled after other articles published in Vaccine.

Overall, then, the paper is interesting. Theoretical framework is weak, and some results create confusion… 

Clarifications made throughout.

The comparison between these two infectious diseases is not consistent because COVID-19 is a big pandemic, Monkeypox has had a low impact and of course emerge differences that have to be better synthetized in a table. These aspects have to be accurately discussed also in terms of limitations of this study. Additionally for COVID-19 there was lockdown and other policy restrictions for Monkeypox not and this has to be also explained to justify the different perceptions and social reaction of mental health.  Moreover, , structure of the paper has to be improved; study design, discussion and presentation of results have to be clarified using suggested comments.

For our dataset, there was no COVID-19 lockdowns, and only at the very end did masks even appear. Within both datasets, there was a novel reaction. Both epidemics were starting. These issues were addressed in the discussion.

If the paper is improved considering that a low relation with the scope of the journal, by using strictly all comments that I will in-depth verify, maybe it can be considered. 

Suggested readings of relevant papers that have to be read and all inserted in the text and references.

We hope we made all suggestions and improved the entire paper.

Round 2

Reviewer 3 Report

I have read thoroughly the revised version of paper.

Now this version of the paper after revision done is OK and provides interesting results for readers.